# A Molecular Dynamics Simulation for Thermal Activation Process in Covalent Bond Dissociation of a Crosslinked Thermosetting Polymer

**DOI:** 10.3390/molecules28062736

**Published:** 2023-03-17

**Authors:** Naoki Yamada, Yutaka Oya, Nobuhiko Kato, Kazuki Mori, Jun Koyanagi

**Affiliations:** 1Department of Materials Science and Technology, Graduate School of Tokyo University of Science, Tokyo 125-8585, Japan; 2Research Institute for Science & Technology, Tokyo University of Science, Tokyo 125-8585, Japan; 3Sience and Engineering Systems Division ITOCHU Techno-Solutions Corporation, Tokyo 105-6950, Japan; 4Department of Materials Science and Technology, Tokyo University of Science, Tokyo 125-8585, Japan

**Keywords:** molecular dynamics, thermosetting resin, mechanical properties, Monte–Carlo method

## Abstract

A novel algorithm for covalent bond dissociation is developed to accurately predict fracture behavior of thermosetting polymers via molecular dynamics simulation. This algorithm is based on the Monte Carlo method that considers the difference in local strain and bond-dissociation energies to reproduce a thermally activated process in a covalent bond dissociation. This study demonstrates the effectiveness of this algorithm in predicting the stress–strain relationship of fully crosslinked thermosetting polymers under uniaxial tensile conditions. Our results indicate that the bond-dissociation energy plays an important role in reproducing the brittle fracture behavior of a thermosetting polymer by affecting the number of covalent bonds that are dissociated simultaneously.

## 1. Introduction

Carbon-fiber-reinforced plastics (CFRPs) have been applied to various structural materials in the aerospace field owing to their high specific strength and specific stiffness [1,2]. As CFRP has a complex inhomogeneous structure made of carbon fiber (reinforcement) and thermosetting polymer (matrix portion), it undergoes various forms of fracture, often initiated as microscopic damage at matrix portions [3,4,5,6]. The thermosetting polymer has a three-dimensional crosslinked structure, and covalent bonds constituting the crosslinks break under mechanical loading. The dissociation of the covalent bonds expands to the macroscopic scale, resulting in the fracture of a CFRP. Therefore, it is necessary to elucidate the mechanisms of microscopic damage to improve the toughness and prevent such fracture behavior of thermosetting polymers. It is difficult to investigate the time evolution of microscopic-scale damages using meso- to macro-scale approaches such as conventional experiments and finite element analysis. This study investigates the time evolutions of covalent bond dissociation and fracture behavior of a thermosetting polymer based on molecular dynamics (MD) simulation, which is essential for elucidating microscopic damage mechanisms of matrix crack and transverse crack in CFRPs. MD simulation has been applied to polymers [7,8,9,10], reinforcements [11,12], and their composites [13,14,15,16,17,18,19,20,21,22,23,24,25], which have quantitatively reproduced thermomechanical properties near equilibrium state such as density, Young’s modulus, and glass transition temperature. Characteristics in the higher-strain region, where covalent bond dissociation is involved, remain challenging to simulate.

In MD simulation, an individual atom in the condensed molecular system is propagated in time based on the (extended) Newton equation, and thermodynamic properties are evaluated as a statistical average of the behavior of all atoms [26]. MD simulations traditionally assume that the topology of the individual molecules does not change during time evolution (such an MD method is hereafter referred to as classic MD). MD simulations using the reactive force field (Reax-FF) have recently attracted significant attention because this method overcomes the limitations associated with classic MD and reproduces phenomena associated with molecular topology changes, namely chemical reaction and microscopic failure [27]. Reax-FF smoothly represents the formation and dissociation of covalent bonds by approximating the bond order as a continuous function of interatomic distance, although it requires significantly more computational resources than classic MD. Numerous studies using Reax-FF have investigated the mechanical properties over a wide range of strains [28,29], which is difficult to achieve with classic MD.

Odegard et al. utilized Reax-FF for the epoxy-based thermosetting polymer [28]. They quantitatively reproduced an elastic response and yielding point under the tensile simulation. Koo et al. simulated the brittle fracture behavior of thermosetting epoxy polymer [29]. They developed new methodology based on Reax-FF with an ultrahigh strain rate approach for the mechanical response over a wide strain range in order to establish both accuracy and numerical efficiency. The resulting maximum strains are in quantitative agreement with the experimental values. Jang et al. investigated the effect of the nanoscale defects on mechanical properties by classic MD with Morse bond potential [30]. In this potential, covalent bonds between atoms are represented by anharmonic potentials, and covalent bond breaking can be approximated as a continuous function of interatomic distance. The results proved that the defect content has a significant effect on the stress–strain response, and realistic fracture behavior can be reproduced by introduction of the nanoscale defects. Konrad et al. obtained the reactive force field that enables us to smoothly describes the formation and dissociation of thermosetting polymers. In the results, the yield strain and the maximum stress are quantitatively obtained [31]. More recently, the bond dissociation has been described based on course-grained MD simulation. Zhao et al. investigated the fracture behavior of double network structures including physical and chemical crosslinked structures [32]. They represented that the depth of the Lennard-Jones potential for the physical network and scission of the chemical network significantly affect the mechanical responses under tensile simulation.

However, to the best of our knowledge, the brittle behavior of thermosetting polymer has not been adequately reproduced, even with bond-order-based MD such as Reax-FF. Although some previous studies successfully captured maximum stress and strain, “the rapid decrease in stress from maximum value”, which is unique to the brittle fracture of thermosetting polymer, has not been realized. One of main factors for this irreproducibility may be originated from the criterion for the covalent bond dissociation. The conventional algorithm uses only the distance between two atoms as a criterion in dealing with covalent bond dissociation, which may be insufficient because the presence or absence of a covalent bond affects not only the atomic pair but also the atoms surrounding them.

In response to the above background, we propose a novel algorithm to represent covalent bond breaking based on the classic MD method. This algorithm reproduces covalent bond dissociation as realistically as possible within the scope of classic MD using the Monte Carlo (MC) method [33], which considers the strain and bond-dissociation energies of interatomic covalent bonds. The energy-based criteria proposed in this study can properly represent information on atoms around the covalent bond via angle, dihedral angle, and Lennard-Jones and Coulomb potentials. To the best of our knowledge, this is the first study to demonstrate the effectiveness of this algorithm for crosslinked epoxy resin via the stress–strain relationships depending on the bond-dissociation energy.

The remainder of this paper is organized as follows. The next section describes the simulation method and simulation system, including the algorithms for covalent bond dissociation. The third section presents the simulation results and discussions. Some future research directions are also suggested. Finally, the results are concluded.

## 2. Simulation Method

In this study, we first create a crosslinked structure for epoxy resin, which is representative of the thermosetting polymer for structural material. Subsequently, the resulting crosslinked structure is subjected to uniaxial elongational loading, accounting for the dissociation of covalent bonds. In this section, we present the details of the molecular models, simulation conditions, and protocols used to realize these MD simulations. All simulations are conducted using the GROMACS software [34].

### 2.1. Molecular Modeling and Curing Simulation

Epoxy resins form crosslinked structures through chemical reactions between epoxy groups in the base resin and functional groups in the curing agent. In this study, bisphenol A diglycidyl ether (DGEBA) and ethylenediamine (EDA) are selected as the base resin and curing agent, respectively. The details of these molecular structures are shown in Figure 1a. DGEBA has epoxy groups, and EDA has primary amine groups (-NH2) at both ends of their backbone. A primary amine turns into a secondary amine (-NH) through a reaction with an epoxy group. A secondary amine reacts with an epoxy group again to form a tertiary amine (-N). The branching structures obtained in these sequential reactions are linked together to form a crosslinked structure extending over the entire system. In this study, DGEBA and EDA are mixed in an epoxy/amine stoichiometric ratio to achieve total number of atoms almost 2000. The equilibrium state of the DGEBA/EDA mixture is first obtained by relaxation calculations under the *NPT* ensemble (temperature *T* = 300 K and pressure *P* = 1 atm). Using this equilibrated system, the crosslinked structure is created via chemical reaction calculations. To reproduce the accurate molecular structures and their interactions, an all-atom optimized potential for liquid simulation (OPLS-AA) force field is employed [35]. For the electrostatic potential charge, density functional calculations are conducted under the condition of B3LYP/6-31G (Hamiltonian/basis set) [36,37].

In the chemical reaction calculations, we adopt the reaction algorithms based on the distance-based criteria, i.e., both functional groups react if the carbon atom of the epoxy group approaches the nitrogen atom of the amine group within a certain distance (6.0 Å). The crosslinked structure is finally obtained by relaxation calculation under *NPT* ensemble (*T* = 300 K, and *P* = 1 atm) combined with this reaction algorithm. Figure 1b shows a snapshot of the calculated crosslinked structure. It should be noted that the chemical reaction of thermosetting resins is a thermal activation process. Similar to previous studies, it may be better to consider the reaction probability based on the Arrhenius equation during the chemical reactions [38,39,40,41,42,43]. However, the chemical reaction of the distance-based criterion used in this study is also supported by many previous studies because it guarantees quantitative accuracy in thermomechanical properties [44,45,46,47,48,49,50,51].

### 2.2. Uniaxial Tensile Simulation Considering Covalent Bond Dissociation

Here, detailed simulation algorithms and conditions are introduced for the uniaxial tensile test incorporating the covalent bond dissociation in a crosslinked epoxy polymer. Figure 2 shows a flowchart of this tensile test. This flowchart is roughly classified into three steps: tensile calculation of the entire system, judgment of covalent bond dissociation with the aid of an MC method, and short time structural relaxation. Finally, this calculation is terminated when the system reaches the predetermined strain. In the tensile test, the system is deformed for 1ps at a strain rate of 3.86 × 10^9^/s in the z-direction, while keeping the size of the cross section in the x–y directions fixed. After executing MC method for covalent bond dissociation, the system is relaxed for 1 ps under NVT ensemble conditions to stabilize the system by removing strong forces due to the discontinuous change in the potential energy in the covalent bond dissociation. The protocol in the MC step for the bond dissociation, which is between the tensile and relaxation steps, is described as follows.

The bond-dissociation process is a thermally activated process as well as a crosslinking reaction. To illustrate the realization of this process via our proposed algorithm, Figure 3 represents the potential energy diagram for the bond-dissociation coordinate. First, strain energy ($E_{\mathrm{strain}}$) accumulates in each interatomic covalent bond due to the tensile deformation of the entire system. The strain energy is represented by the difference in the total potential energy before and after bond dissociation. The covalent bond is then broken when the activation energy, which represents the energy difference between this strain energy and the bond-dissociation energy ($E_{\mathrm{BD}}$), is exceeded by thermal fluctuations. To realize such a thermally activated process via MD simulation, we introduce the following three steps using the MC method. These steps correspond to the right-hand side of the flowchart in Figure 2.

Step 1. Select all covalent bonds whose lengths are greater than the bond length criteria $r_{0}$. In this study, this criterion is determined as 0.5–0.7 σ, where σ is the distance at which the Lennard-Jones potential is the smallest in the carbon-to-carbon bond. At around 0.5 σ, the covalent bond starts to dissociate via the preliminary tensile simulation without considering $E_{\mathrm{BD}}$.

Step 2. Evaluate the strain energy for all bonds selected in Step 1. The strain energy of a single bond is expressed as the difference in the total potential energy with and without the bond. First, the total potential energy before dissociating a bond is measured ($E_{\mathrm{before}}$). Then, one of the selected bonds is virtually dissociated, the OPLS-AA forcefield is reassigned for the molecule, and the total potential energy is measured ($E_{\mathrm{after}}$). The strain energy for the single bond is evaluated as this potential energy difference, i.e., $E_{\mathrm{strain}}=E_{\mathrm{before}}-E_{\mathrm{after}}$.

Step 3. The bond dissociation is determined for all covalent bonds selected in Step 1 via the MC method, accounting for $E_{\mathrm{strain}}$ evaluated in Step 2. In the MC method, the bond is dissociated if the strain energy is greater than $E_{\mathrm{BD}}$. If it is equal or smaller, the bond is dissociated according to the probability (*k*) estimated using the Arrhenius equation as follows:(1)$$k=A\;\exp\left(-\frac{E_{\mathrm{BD}}-E_{\mathrm{strain}}}{RT}\right)$$
where R is the gas constant, and *T* is the local temperature. *A* is a frequent factor set to 1 in this study. The probability evaluated by Equation (1) is compared with a uniform random number (*r*) in the range of 0–1: adopt the virtual dissociation in Step 2 if r<k; otherwise, reject it.

It should be noted that atoms with different types of bonding atoms are generally distinguished as different atomic types, even if they are of the same atomic species. For example, a carbon atom in methane and a carbon atom in a benzene are considered different types of atoms. Therefore, it is better to define a new atomic type after the bond dissociation. However, for simplicity, this study assumes that the atom type does not change before and after bond dissociation.

In this study, the uniaxial tensile simulations are performed with changing $r_{0}$ and $E_{\mathrm{BD}}$ as parameters. Although $E_{\mathrm{BD}}$ depends on the covalent bond type, e.g., C–N and C–C are different in $E_{\mathrm{BD}}$, this study used the same value for simplicity. For stress–strain diagram, the strain is evaluated by $\left(L_{z}\left(t\right)-L_{z}\left(t=0\right)\right)/L_{z}\left(t=0\right)$, where $L_{z}\left(t\right)$ is the length of the system in the tensile direction (*z*-direction) at time *t*. Based on the virial theorem, the stress $\left(\sigma \equiv \sigma_{zz}\left(t\right)\right)$ at time *t* is expressed by $\partial \left(K\left(t\right)+\varphi \left(t\right)\right)/\partial \varepsilon_{zz}\left(t\right)/V\left(t\right)$, where K(t) and φ(t) are kinetic energy and potential energy of the system, V is the system volume, and 〈⋯〉 represents the statistical averaging operation. To ensure statistical correctness, three independent simulations were performed for each parameter. The results show that the stress–strain diagrams of the three samples are almost identical in their shape, and characteristic stress and strain values, e.g., the errors in the maximum strain and stress, are within 10%, respectively. Therefore, in this study, the sample with intermediate values on the stress–strain diagram was selected as representative.

## 3. Results and Discussion

Figure 4a shows the stress–strain curves depending on $r_{0}$ for $E_{\mathrm{BD}}=0$, where each covalent bond dissociates upon reaching the criterion length $r_{0}$. For all curves, the stress monotonically increases with the system deformation, and suddenly decreases due to the bond dissociation event. This stress increase and decrease in stress are repeated three, four, or five times, and finally, the stress reaches zero, i.e., the system is fractured. The slope of the stress–strain graph decreases with the bond dissociation events because the number of bonds supporting the load decreases. Consequently, the stress gradually decreases from its maximum value as the system deforms, as reported previously [52,53]. This figure also shows that $r_{0}$ does not change the shape of the stress–strain curve, although the maximum stress and strain are significantly affected. Notably, the stress changes discretely owing to the small system size. As the number of atoms inside the system increases, the change in the stress becomes smoother even at the dissociation events.

Figure 4b shows the stress–strain curves depending on $E_{\mathrm{BD}}$ for $r_{0}=0.55\sigma$. By introducing $E_{\mathrm{BD}}$, the following two changes appear in the stress–strain curve. First, the maximum stress and its strain increase with $E_{\mathrm{BD}}$. This occurs because the larger strain energy $E_{\mathrm{strain}}$ is required to overcome the larger $E_{\mathrm{BD}}$ for the bond dissociation. Second, the larger $E_{\mathrm{BD}}$ results in a smaller difference in the strain at the maximum stress and material fracture because of the larger single stress reduction. In particular for $E_{\mathrm{BD}}=400\;\mathrm{kJ}/\mathrm{mol}$, more than 80% of the maximum stress is reduced at a strain of around 1.7. Thus, brittle-like behavior inherent to thermosetting polymers is reproduced.

To understand the fracture behavior depending on $E_{\mathrm{BD}}$, Figure 5a,b show the number of dissociated covalent bonds (*N*) with respect to the strain (*ε*) and $E_{\mathrm{BD}}$, respectively. These figures show the results of the three characteristics associated with *N*. First, the strong positive correlations between *N* and *ε* as well as *N* and $E_{\mathrm{BD}}$ are presented. Second, *N* at the first event increases with $E_{\mathrm{BD}}$. Third, the smaller $E_{\mathrm{BD}}$ increases the bond dissociation event. These three results indicate the following microscopic aspects. For a smaller $E_{\mathrm{BD}}$, the strong load applied to the system is supported by a small number of covalent bonds, leading to the sequential replacement of the loaded bond every time the bond breaks. Consequently, multiple events in the covalent bond dissociation occur. However, in the case of a large $E_{\mathrm{BD}}$, many covalent bonds in the entire system become too unstable to support the strong loads via large deformations. Therefore, many covalent bonds are simultaneously dissociated via thermal fluctuation, causing a large stress reduction.

Figure 6 shows the snapshots of the system during the tensile deformation for (a) $E_{\mathrm{BD}}=0$ and (b)$\;E_{\mathrm{BD}}=400$ (kJ/mol), visualizing the effect of $E_{\mathrm{BD}}$ on the fracture behavior of the covalent bonds. For $E_{\mathrm{BD}}=0$, the covalent bonds that constitute a molecular segment elongated in the tensile direction are successively dissociated with the deformation from *ε* = 0.5 to 2.0. For $E_{\mathrm{BD}}=400$, the covalent bonds in tension are widely distributed inside the system, and several of these bonds are simultaneously broken between *ε* = 1.5 and 2.0. Fragments of molecules not included in the crosslinked structure are also found at *ε* = 2.0. It should be noted that $E_{\mathrm{BD}}$ of the covalent bonds in the backbone of the DGEBA/EDA reactant is in the range of 270–380 kJ/mol, according to the calculations by ALFABET, a software based on quantum calculations and machine learning techniques [54,55,56]. $E_{\mathrm{BD}}$ used in this study is believed to be adequate for a practical system.

The following inferences can be drawn from the results of this study. When $E_{\mathrm{BD}}$ is small, the stress gradually decreases from the first peak because the crosslinked structure is broken sequentially. In contrast, for a larger $E_{\mathrm{BD}}$, the crosslinked structure is simultaneously broken throughout the system, causing a drastic reduction in the stress and brittle behavior of the material.

The stress–strain curve for the brittle fracture behavior can be reproduced by introducing the $E_{\mathrm{strain}}$ and $E_{\mathrm{BD}}$. However, the maximum stress and strain are significantly different from those determined experimentally. One of the main reasons for the discrepancy between the simulation and experimental results may be the inaccurate treatment during the electrostatic interactions. In this study, the point charge for each atom does not change during the MD simulation. However, the electrostatic field around a molecule depends on molecular conformation. In particular, the electrostatic field significantly changes in response to the molecular topological change associated with the chemical reaction and covalent bond fracture. Owing to the covalent bond dissociation, unpaired electrons are usually generated, resulting in the destabilization of separated molecules. Such a destabilization is expected to affect the proposed thermally activated process. The point charge should be redefined to improve our protocols.

In the future, the stress–strain curve over a wide range of system deformations will be quantitatively reproduced by scaling up the system, refining the electrostatic interaction, and other parameters such as $r_{0}$, $E_{\mathrm{BD}}$, and the frequency factor in Equation (1). Furthermore, failure mechanisms for various loading conditions, including the cyclic loading for the fatigue failure, will be elucidated on micro to macro scales by combining this approach with coarse-grained simulations such as dissipative particle dynamics [57], density functional theory [58], and the finite element method [59].

## 4. Conclusions

In this study, we developed a novel algorithm that can represent a thermally activated process during covalent bond dissociation by combining classic MD simulation with the MC method. Many previous studies based on classic MD and bond-order-based MD have considered only covalent bond length when determining bond dissociation. However, the angles between the neighboring three and four atoms are also important for determining the strain energy of a covalent bond. For the first time, in this study, the strain energy accumulated in a single bond is represented by the sum of all potential energy contributions, such as bond, angle, dihedral angle, and Lennard-Jones and Coulomb potentials. Another novel aspect of this study is the introduction of the bond dissociation energy ($E_{\mathrm{BD}}$), which the strain energy exceeds during bond dissociation.

This algorithm is applied to the uniaxial tensile simulations for the crosslinked structure obtained by the chemical reaction between DGEBA and EDA, leading to the following results. When $E_{\mathrm{BD}}$ is small, the stress decreases gradually from its maximum value because the covalent bonds are sequentially broken. However, many covalent bonds are simultaneously broken for a large $E_{\mathrm{BD}}$, resulting in a drastic reduction in the stress. These results indicate that covalent bond dissociation as a thermal activation process strongly influences the brittle fracture behavior of thermosetting polymers.

## Figures and Tables

**Figure 1 molecules-28-02736-f001:**
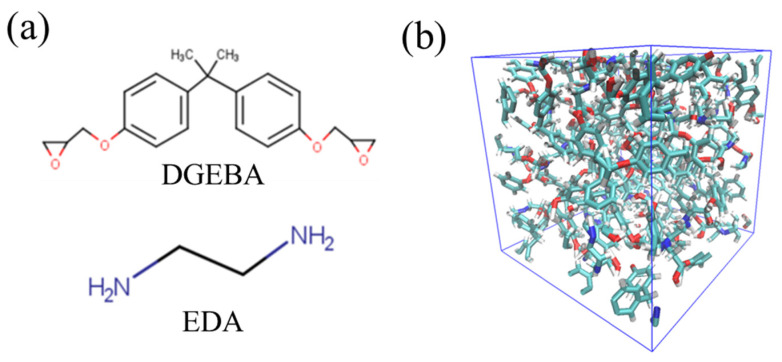
(**a**) Molecular structures of DGEBA (upper) and EDA (lower), and (**b**) snapshot of the crosslinked structure.

**Figure 2 molecules-28-02736-f002:**
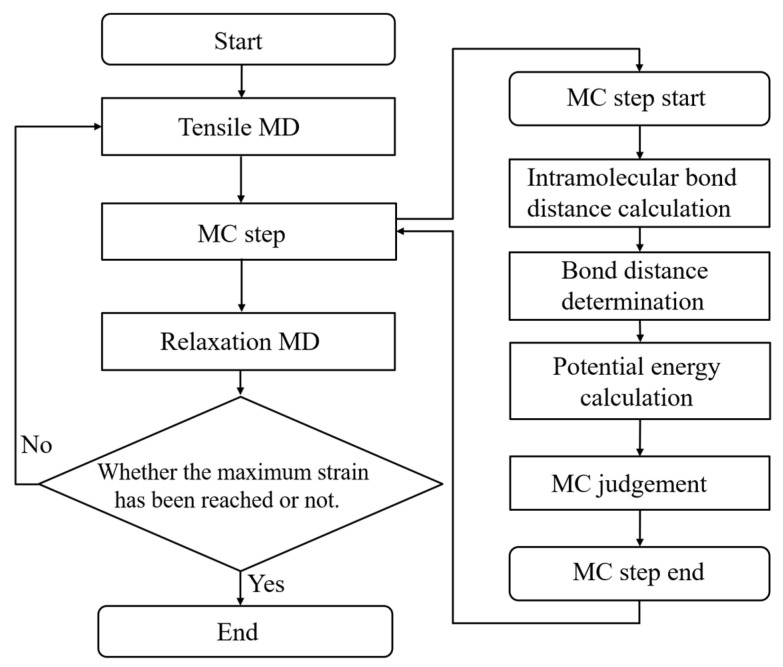
Flow chart of the tensile simulation. Right-hand part shows the detailed algorithms in the MC step for reproducing the covalent bond dissociation.

**Figure 3 molecules-28-02736-f003:**
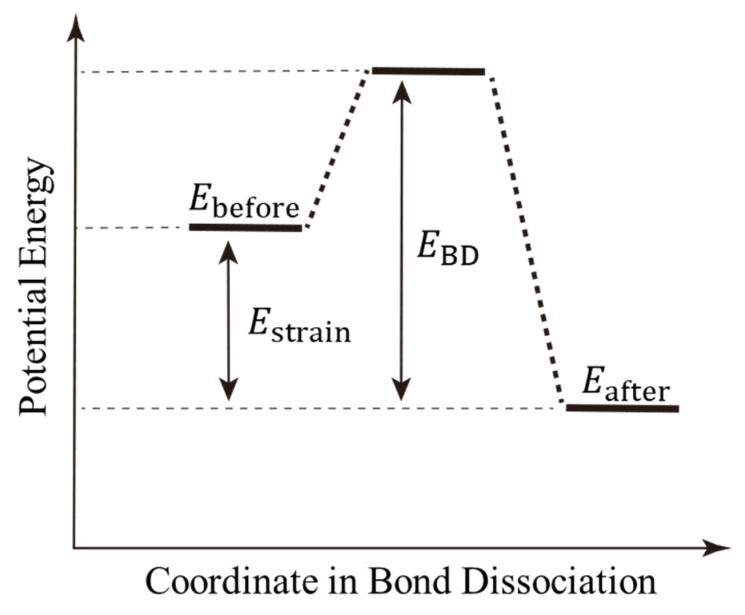
Schematic of the potential energy changes for the bond-dissociation process. $E_{\mathrm{before}}$ and $E_{\mathrm{after}}$ are the potential energies before and after bond dissociation, respectively. $E_{\mathrm{strain}}$ and $E_{\mathrm{BD}}$ are the strain and bond-dissociation energies, respectively.

**Figure 4 molecules-28-02736-f004:**
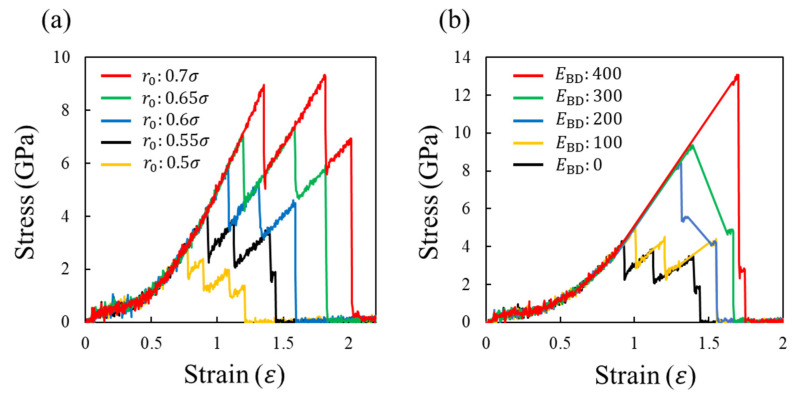
(**a**) Stress–strain curves depending on r_0 at E_BD = 0 and (**b**) those depending on E_BD at r_0 = 0.55σ. There figures have different scales in the horizontal and vertical axis.

**Figure 5 molecules-28-02736-f005:**
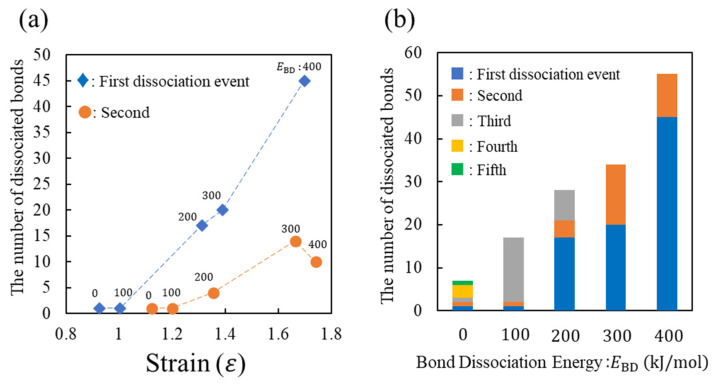
Number of dissociated bonds with respect to (**a**) strain and (**b**) E_BD, whose results correspond to Figure 4b. In subfigure (**a**), the number of the first and second dissociated bonds is represented by blue diamond (♦) and orange circle (●), respectively. The corresponding value of E_BD is listed near each datum. In subfigure (**b**), numbers from the first to fifth dissociation events are represented in blue, orange, grey, yellow, and green, respectively.

**Figure 6 molecules-28-02736-f006:**
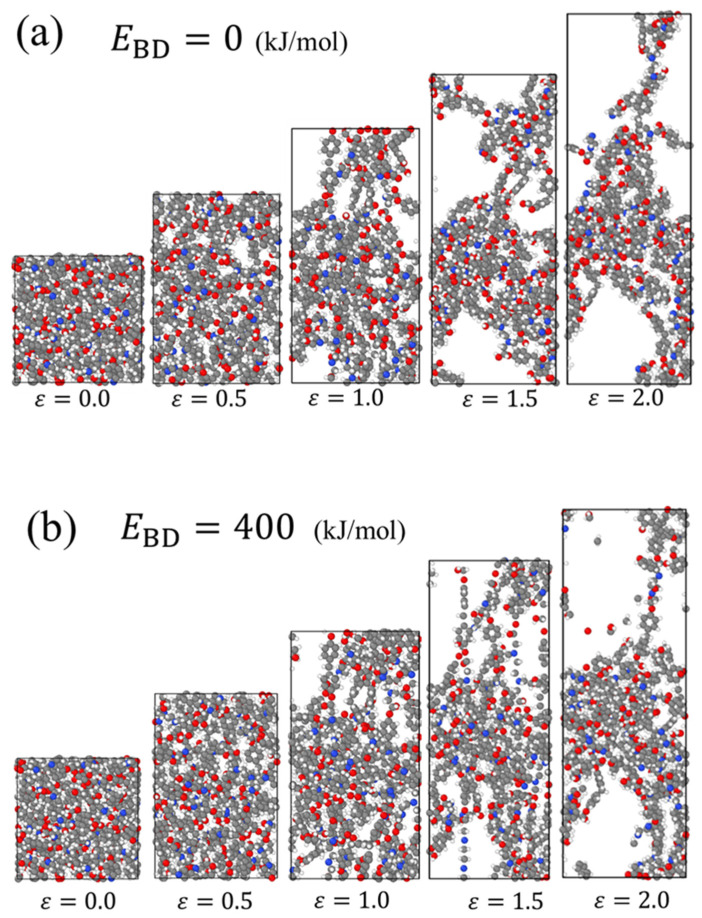
Snapshots of the system during uniaxial tensile simulation for (**a**) $E_{\mathrm{BD}}=0$, and (**b**) $E_{\mathrm{BD}}=400$ (kJ/mol).

## Data Availability

Not applicable.

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
