# Peer review of "A Molecular Dynamics Simulation for Thermal Activation Process in Covalent Bond Dissociation of a Crosslinked Thermosetting Polymer"

_molecules, 2023, doi:10.3390/molecules28062736_

Round 1
Reviewer 1 Report
Overall evaluation: In this study, the covalent bond dissociation of fracture behavior in thermosetting polymers was predict by a novel algorithm. The thermally activated process during covalent bond dissociation was represented by combining traditional MD simulation with MC method. Meanwhile, the difference in local strain and bond-dissociation energies was considered. Finally, the effect of bond dissociation energy in brittle fracture behavior was investigated, which played an important role. However, more work should be added in order to completely sturdy. Therefore, minor revision is recommended, and comments are listed below:
1. The fracture process in the crosslinked thermosetting polymer should be studied. The others work (Polymer, Volume 244, 23 March 2022, 124670) could be referred.
2. The main problem statement and justification for the research has not been clearly stated. Meanwhile, the significance of investigate covalent bond dissociation was not clear.
3. In general, there is a lack of explanation of replicates and statistical methods used in the study. Also, there are few explanations of the rationale for the study design.
4. Try to set the problem discussed in this paper in clearer, and write to define the problem.
5. The other literature could be compared in Introduction, and its work and shortcomings need to be pointed out.
6. The size of Fig. 2 and 3. should be shorten. Otherwise, the text size is larger than the text size.
7. Conclusions sections should be amended briefly.
Reviewer 2 Report
Authors present a combined MD/MC computational approach to the simulation of brittle behaviour and failure of a thermosetting, crosslinked, polymer due to mechanical stress. In particular, for a given applied strain, the covalent bonds which are broken are evaluated according to a MC statistical procedure, to model the thermal activation of the bond breaking process. Different values of the bond-dissociation energy parameter are tested, predicting different outcomes (gradual or sudden bonds breaking) of the applied mechanical tension.
I think that the methodologies and the results are well presented. Therefore, I suggest publication in Molecules journal after the following minor points have been corrected or clarified:
1) In order to evaluate the strain energy, are the atom types (for example those for the dissociated atoms) the same before and after bond breaking? Did the MM potential energy differ only for one bond (stretching) term? Authors could better clarify these aspects in the strain energy calculation.
2) The “stress” and “strain” quantities should be better defined in the Manuscript before to employ them in Results section.
3) Please correct the following sentences:
… an individual atom in the condensed molecular system is temporally developed… (propagated in time)
…to achieve a total atomic number of almost 2000. (total number of atoms?)
…strain rate of 3.86×109/s… (109)
…potential energy diagram for the bond dissociate coordinate (dissociation).
